# Antimicrobial Resistance Surveillance of Tigecycline-Resistant Strains Isolated from Herbivores in Northwest China

**DOI:** 10.3390/microorganisms10122432

**Published:** 2022-12-08

**Authors:** Yongfeng Yu, Changchun Shao, Xiaowei Gong, Heng Quan, Donghui Liu, Qiwei Chen, Yuefeng Chu

**Affiliations:** 1State Key Laboratory of Veterinary Etiological Biology, Lanzhou Veterinary Research Institute, Chinese Academy of Agricultural Sciences, Lanzhou 730046, China; 2Lanzhou Institute for Food and Drug Control, Lanzhou 730050, China; 3College of Veterinary Medicine, Gansu Agricultural University, Lanzhou 730070, China

**Keywords:** genome sequence, resistome, virulence factors, *Escherichia coli*, herbivores, northwest China

## Abstract

There is no doubt that antimicrobial resistance (AMR) is a global threat to public health and safety, regardless of whether it’s caused by people or natural transmission. This study aimed to investigate the genetic characteristics and variations of tigecycline-resistant Gram-negative isolates from herbivores in northwest China. In this study, a total of 300 samples were collected from various provinces in northwest China, and 11 strains (3.67%) of tigecycline-resistant bacteria were obtained. In addition, bacterial identification and antibiotic susceptibility testing against 14 antibiotics were performed. All isolates were multiple drug-resistant (MDR) and resistant to more than three kinds of antibiotics. Using an Illumina MiSeq platform, 11 tigecycline-resistant isolates were sequenced using whole genome sequencing (WGS). The assembled draft genomes were annotated, and then sequences were blasted against the AMR gene database and virulence factor database. Several resistance genes mediating drug resistance were detected by WGS, including fluoroquinolone resistance genes (*gyrA_S83L*, *gyrA_D87N*, *S83L*, *parC_S80I,* and *gyrB_S463A*), fosfomycin resistance genes (*GlpT_E448K* and *UhpT_E350Q*), beta-lactam resistance genes (*FtsI_D350N* and *S357N*), and the tigecycline resistance gene (*tetR N/A*). Furthermore, there were five kinds of chromosomally encoded genetic systems that confer MDR (*MarR_Y137H*, *G103S*, *MarR_N/A*, *SoxR_N/A*, *SoxS_N/A*, *AcrR N/A,* and *MexZ_K127E*). A comprehensive analysis of MDR strains derived from WGS was used to detect variable antimicrobial resistance genes and their precise mechanisms of resistance. In addition, we found a novel ST type of *Escherichia coli* (ST13667) and a newly discovered point mutation (*K127E*) in the *MexZ* gene of *Pseudomonas aeruginosa*. WGS plays a crucial role in AMR control, prevention strategies, as well as multifaceted intervention strategies.

## 1. Introduction

One of the biggest threats to global health is antimicrobial resistance, affecting the environment, animals, and humans [1]. Gram-negative bacteria such as *Escherichia coli* (*E. coli*), *Klebsiella pneumoniae (K. pneumoniae*), *Pseudomonas aeruginosa* (*P. aeruginosa*), and *Salmonella typhimurium* (*S. Typhimurium*) are important zoonosis pathogens [2]. With the extensive application of antibiotics in livestock breeding and treatment, the rapid increase in the prevalence of extensively drug-resistant (XDR) Gram-negative bacteria, particularly carbapenem-resistant *Enterobacteriaceae* and *Acinetobacter* spp., have affected the efficacy of carbapenems. For example, *Enterobacter*, as an indicator of the prevalence of Gram-negative bacteria, is a rich antibiotic resistance gene pool and a mobile center for drug resistance gene exchange [3,4]. Therefore, in many investigations of Gram-negative drug-resistant bacteria, many strains were found together with *E. coli*. Additionally, *K. pneumoniae*, *P. aeruginosa,* and *S. Typhimurium* have been mentioned and found to contain multiple drug-resistant (MDR) strains in previous reports [5,6]. In brief, the cross-infection of multiple Gram-negative drug-resistant bacteria carrying different drug-resistant genes have brought significant challenges to clinical prevention and the treatment process [7]. Therefore, it is extremely critical to distinguish and characterize the drug-resistance characteristics of different drug-resistant strains.

At present, MCR-type colistin-resistant strains are widely reported in *Enterobacteriaceae*, and tigecycline is one of the last resort antibiotics for treating these superbugs [8,9]. Originally derived from tetracycline, tigecycline was designed to overcome tetracycline resistance’s common mechanism [10]. Tigecycline inhibits bacterial growth by binding to the 30S ribosome and blocking the entry of tRNA, thus preventing protein synthesis. Furthermore, tigecycline escapes tetracycline resistance mechanisms due to its different binding orientation [11]. Tigecycline is regarded as a last-line antibiotic against infections caused by MDR or XDR bacterial pathogens, so long-term use of tigecycline is not recommended. However, several cases of tigecycline resistance have been reported in the scientific community since tigecycline was first used clinically [12,13,14]. Most cases of tigecycline resistance were attributed to one or more of the following mechanisms: mutations within the ribosomal binding site, acquisition of mobile genetic elements carrying tetracycline-specific resistance genes, and/or chromosomal mutations leading to the increased expression of intrinsic resistance mechanisms [15,16]. Therefore, when tetracycline-resistant strains become prevalent, we will face the dilemma that there is no effective antibiotic available.

Furthermore, as the natural pasture of animal husbandry in China, the northwest region has a unique advantage in herbivore breeding. However, the prevalence of drug-resistant strains has brought serious economic losses to the aquaculture industry in this area [17,18,19]. So far, there has been no investigation of tigecycline-resistant strains in northwest China, and no whole-genome sequencing (WGS) analysis of tigecycline isolates. Therefore, it is particularly crucial to analyze the drug resistance mechanism and molecular characteristics of these tigecycline-resistant strains to provide a theoretical basis and new programs for clinical treatment and prevention. In this study, fresh stool samples were collected from herbivores of different varieties under different farming environments in northwest China from June 2021 to May 2022, and tigecycline-resistant strains were analyzed in the samples to address the inadequacy of previous studies on the drug-resistant bacteria of animal origin in this area. We performed WGS on the tigecycline-resistant isolates (including *E. coli*, *K. pneumoniae*, *P. aeruginosa*, and *S. Typhimurium*) to uncover the prevalence and genetic diversity of tigecycline-resistant strains derived from animals.

## 2. Materials and Methods

### 2.1. Sample Collection and Bacterial Isolates

In the present study, 300 stools were sampled in eight study plots located on 12 large-scale farms in northwest China from June 2021 to May 2022. We took stool samples from 150 cattle and 150 sheep in various breeding modes, including males, females, and young animals (Figure 1 and Table 1). We incubated 0.5 g of feces in 5 mL of Luria-Bertani (LB) to enrich bacteria for 6 h. We screened the tigecycline-resistant strains on an LB plate with 2 μg/mL tigecycline [20]. Species and genera of the screened single strains were identified by 16S sequencing and preserved in 60% glycerol.

### 2.2. Antimicrobial Susceptibility

According to Clinical and Laboratory Standards Institute (CLSI) guidelines [21], the minimal inhibitory concentration (MIC) of the isolated strains (tigecycline-resistant strains, MIC ≥ 2 μg/mL) to 14 antibiotics was tested by the double broth microdilution method. The MIC values of all strains were determined on three separate occasions. *E. coli* ATCC 25922 was used as quality control in all drug sensitivity tests. We interpreted the antibiotic susceptibility results based on the CLSI, 2020 breakpoints. Tests were conducted on a panel of antimicrobial compounds, including amoxicillin/clavulanate potassium (AMC, 4/2–128/64 μg/mL), ampicillin (AMP, 2–128 μg/mL), meropenem (MEM, 0.5–16 μg/mL), cefotaxime (CTX, 0.06–8 μg/mL), gentamicin (GEN, 0.25–32 μg/mL), ceftiofur (EFT, 0.25–32 μg/mL), amikacin (AMK, 2–64 μg/mL), fosfomycin (FOS, 0.25–32 μg/mL), colistin (CL, 0.125–8 μg/mL), sulfamethoxazole (SXT, 9.5–304 μg/mL), ciprofloxacin (CIP, 0.06–8 μg/mL), tetracycline (TET, 0.25–64 μg/mL), tigecycline (TIG, 0.25–32 μg/mL), and florfenicol (FFC, 2–128 μg/mL) [22]. All antibiotics were purchased from Solarbio (Beijing, China).

### 2.3. Whole-Genome Sequencing

Using a commercially available bacterial genomic DNA isolation kit (Generay, China), DNA was obtained from isolates that displayed tigecycline resistance according to the manufacturer’s instructions. A NanoDrop 2000 spectrophotometer (Thermo Fisher, Waltham, MA, USA) was used to measure the DNA concentration in the extracted samples. The genome samples were interrupted, and the sticky ends were repaired into flat ends by T4 DNA Polymerase, Klenow DNA Polymerase, and T4 PNK. By adding a base ‘A’ at the 3’ terminal, the DNA fragment could be connected to a special junction with an ‘A’ base at the 3’ terminal. DNA fragments of the target size were selected by the magnetic bead method and the high-fidelity PCR enzyme-enriched DNA-seq library. Finally, qualified libraries were fully sequenced by whole genome paired-end sequencing (Illumina, San Diego, CA, USA), thus producing 150-bp paired-end reads (PE 150).

### 2.4. Identification of Antimicrobial Resistant Genes, Multilocus Sequence Typing Analysis, and Virulence Factors

Each draft genome was screened for genes associated with AMR. As a reference for determining drug-resistance genes in isolates, the most updated AMR gene database was downloaded from the NCBI National Database of Antibiotic-Resistant Organisms (accessed on 16 August 2022) [23]. A database of virulence factors and the Virulence Finder 2.0 software were used to predict virulence factors [24,25]. PubMLST (https://pubmlst.org/; accessed on 20 September 2022) was used to perform multi-locus sequence typing (MLST) of assembled bacterial genomes [26].

### 2.5. Statistical Analysis

Sangerbox software (v1.1.3) (http://vip.sangerbox.com; accessed on 3 August 2022) was used to make heat maps of drug resistance characteristics, drug resistance genes, and the virulence factors of isolates. In the cluster analysis of drug-resistant characteristics, the presence of the above resistance phenotype received a score of 1, the intermediate received a score of 0, and the susceptibility received a score of −1. In the cluster analysis of drug-resistant genes and virulence factors, the existence received a score of 1, and the nonexistence received a score of 0.

## 3. Results

### 3.1. Tigecycline Resistant Isolates

The specific sampling time, sampling volume, prevalence, and geographical location are shown in Table 1 and Figure 1. A total of 11 tigecycline-resistant strains were isolated and identified from 300 stool samples collected in northwest China, with an isolation rate of 3.67%. Four *E. coli* strains resistant to tigecycline were identified from 60 samples collected from Shaanxi (6.67%). Two strains of *E. coli* and one strain of *K. pneumoniae* resistant to tigecycline were identified from 60 samples collected from Xinjiang (5.0%). One *E. coli* strain resistant to tigecycline was identified from 30 samples collected in Sichuan (3.33%). Two strains resistant to tigecycline, *S. Typhimurium* and *P. aeruginosa,* were detected from 90 samples collected in Gansu (2.22%). One *K. pneumoniae* strain resistant to tigecycline was identified from 30 samples collected from Qinghai (3.33%), and no tigecycline-resistant isolate was identified from 30 samples collected from Tibet (0%). It is clear from this data that Shaanxi province had the highest isolation rate, while Tibet had no tigecycline-resistant strains. Meanwhile, tigecycline-resistant isolates from Xinjiang, Qinghai, and Gansu provinces showed a diversity of strains rather than a predominance of *E. coli*. 

### 3.2. Antibiotic Sensitivity Test

The sensitivity test results of 11 tigecycline-resistant strains to 14 antibiotics are shown in Figure 2 and Table 2. *E. coli* and *K. pneumoniae* strains have the most serious drug resistance and were the most prevalent tigecycline-resistant strains in the region. In terms of the MIC distribution (Table 2), the MIC values of the antibiotics tetracycline, ampicillin, and sulfamethoxazole were significantly higher. In addition, all isolates were sensitive to ceftiofur, meropenem, gentamicin, and colistin; only SX-3E-11 was extremely sensitive to florfenicol. This may suggest potential drug delivery strategies in these areas. In addition, SX-2E-03 and SX-1E-06 had the same drug resistance spectrum, and they were isolated from the same farm. Other isolates showed different drug resistance profiles, even those isolated from different farms in the same area (SX-3E-11 and SX-1E-06). The outcome of the antibiotic resistance pattern is depicted in Figure 2. All tigecycline-resistant isolates could be typed into four different antibiotypes, and among these resistance patterns, profiles number 6 (with 5 isolates) and 7 (with 3 isolates) had the highest frequencies. Among the 11 isolates, only one strain showed resistance to five of the tested antibiotic categories (XJ-1E-02: AMP-CIP-SXT-TIG-TET). Two of these strains showed resistance to eight of the tested antibiotic categories, with two types of resistance, AMP-CTX-AMK-SXT-TIG-TET-FFC-FOS-resistant (LZ-1S-01) and AMP-AMC-AMK-CIP-SXT-TIG-TET-FFC-resistant (LZ-1P-09). The multidrug resistance assay indicated that all tigecycline-resistant strains are common and have diverse and wide AMR spectra. This suggests that the difference in the antimicrobial spectrum of strains may be due to the frequency of the types of antibiotics used by different farms in clinical breeding and treatment.

### 3.3. Whole Genome Sequencing Analysis 

The final assembly of the isolates, based on WGS, ranged from 104 to 187 contigs of >500 bps/sample in *E. coli* isolates with N50 values between 49,186 and 88,174. A total of 127, 155, 152, 104, 187, 129, and 188 contigs, representing 5,154,027; 5,417,711; 5,417,297; 5,071,184; 5,105,116; 5,066,943; and 4,819,548 bases (50.32% and 50.94% G + C ratio; N50 = 181,274; 123,205; 123,205; 143,900; 111,759; 177,264; and 96,022), were obtained from assembled sequences of *E. coli* strains SX-3E-11, SX-2E-03, SX-1E-06, SX-1E-08, SC-1E-03, XJ-2E-05, and XJ-1E-02, respectively. 

The isolates’ final assembly of 81 and 82 contigs of >500 bps/sample in *K. pneumoniae* isolates with N50 values of 234,490 and 221,920, representing 5,574,832 and 6,028,924 bases (57.14% and 55.09% G + C ratio), were obtained from assembled sequences of *K. pneumoniae* strains QH-2K-03 and XJ-1K-13, respectively. The isolates of *P. aeruginosa* LZ-1P-09 and *S. Typhimurium* LZ-1S-01 were assembled by 364 and 31 contigs, respectively, with N50 values of 159,433 and 467,849, representing 31,174,726 and 4,754,898 bases, respectively. The main features of the *E. coli*, *K. pneumoniae*, *P. aeruginosa,* and *S. Typhimurium* genomes are shown in Table 3.

### 3.4. Distribution of Antimicrobial Resistance Genes and Virulence Factors of Isolates

The genomes of all 11 tigecycline-resistant isolates were sequenced, with 12 AMR genes predicted from them (Figure 3A); these include three fluoroquinolone resistance genes (*gyrA_S83L*, *gyrA_D87N*, *S83L*, *parC_S80I,* and *gyrB_S463A*), two fosfomycin resistance genes (*GlpT_E448K* and *UhpT_E350Q*), one beta-lactam resistance gene (*FtsI_D350N*, *S357N*), one tigecycline resistance gene (*tetR_N/A*), and five chromosomally encoded genetic systems that confer MDR (*MarR_Y137H*, *G103S*, *MarR_N/A*, *SoxR/SoxS_N/A*, *AcrR_N/A,* and *MexZ_K127E).* In conjunction with the drug-resistance profiles of the isolates, it can be seen that the distribution of drug-resistance genes is highly consistent with the drug-resistance profiles of the isolates as a whole. It also confirms the correlation between the presence of these resistance genes and resistance phenotypes.

A total of 230 virulence factors were predicted from 11 tigecycline-resistant isolates (Figure 3B,C). All *E. coli* isolates possess 13 virulence factors, including *entA*, *aslA*, *espL1*, etc. In addition, *Pic* exists only in SX-3E-11, *cseA* and *espC* exist only in SX-1E-08, *faeC-J* exists only in SC-1E-03, and *espX6*, *f17d-C,* and *f17d-D* only exist in XJ-2E-05. Furthermore, all *E. coli*, *K. pneumoniae,* and *S. Typhimurium* isolates contain the *entE*, *entF*, *fepA*, *fepG,* and *ompA* genes. Among the isolates of *K. pneumoniae*, *exeF* and *fleQ/flrC* only existed in QH-2K-03, while *clfA* and *htpB* only existed in XJ-1K-13. In addition, some isolates from the same region have similar virulence profiles (SX-2E-03, SX-1E-06, and SX-1E-08), while others have different virulence profiles (XJ-1E-02 and XJ-2E-05). Of note, we found that LZ-1S-09 carried a large number of virulence genes (Figure 3C). It may be due to *Salmonella* serovars containing large, low-copy-number plasmids carrying antibiotic resistance genes or virulence genes. As a result of the presence of animals from various regions on the farm, there may be some differences in their behavior.

### 3.5. Multilocus Sequence Typing (MLST)

The results of MLST showed that 7 *E. coli* isolates belonged to 6 kinds of sequence typing (ST). In particular, ST13667 (SC-1E-03; *adK*, 6; *adK*, 4; *gyrB*, 1323; *icd*,1; *mdh*, 9; *purA*, 2; and *recA*, 7) is a new kind of ST we found, indicating that the *E. coli* isolates in this study are highly diverse. *K. pneumoniae* isolates QH-2K-03 and XJ-1K-13 belong to ST999 and ST65, respectively. LZ-1P-09 of *P. aeruginosa* and LZ-1S-01 of *S. Typhimurium* belong, respectively, to ST68 and ST92 (Table 4).

## 4. Discussion

Antimicrobial resistance is one of the greatest threats to human health in the 21st century, especially with regard to zoonotic pathogens. *E. coli*, *K. pneumoniae*, *S. Typhimurium*, and *P. aeruginosa* are significant zoonotic pathogens that cause a wide range of clinical diseases [2]. Tigecycline is an important drug for the treatment of drug-resistant strains in the clinic, and it is the last line of defense for the treatment of bacterial infection [27]. We present a study in which we first identified the presence of tigecycline-resistant strains in northwest China and then analyzed the drug resistance as well as the WGS of the isolates. Furthermore, this study found that herbivores in northwest China were relatively low in carrying tigecycline-resistant bacteria, which is an interesting finding. Antibiotic resistance genes were widely distributed in isolates, including fluoroquinolone resistance, fosfomycin resistance, and other genes endowed with resistance to β-lactamase, fosfomycin, aminoglycosides, sulfonamides, quinolones, tetracycline and chloramphenicol, and several chromosomally encoded genetic systems that confer MDR. All isolates except LZ-1P-09 were MDR phenotypes that carried at least one β-lactamase gene and the MICs of carbapenem antibiotics supported the presence of resistance genes of these antibiotics. Only one drug resistance gene, *MexZ,* was detected in the *P. aeruginosa* isolate LZ-1P-09, and *MexZ* is the main reason for *P. aeruginosa*’s natural resistance to tigecycline [28]. By comparing the protein sequence of the gene, we found that there was a mutation form *K127E* in *MexZ* which had not been previously reported. In addition, it is interesting that the fosfomycin resistance gene in *E. coli* and *S. Typhimurium* is *GlpT*, and that in *K. pneumoniae* is *UhpT*. Although we screened the isolates by adding tigecycline to the culture medium, only the *tetR* gene was detected. *TetR* is the repressor of the tetracycline resistance element, wherein its N-terminal region forms a helix-turn-helix structure and binds DNA. The binding of tetracycline to *tetR* reduces the repressor affinity for the tetracycline resistance gene (*tetA*) promoter operator sites [29,30]. Therefore, we believe that the reason for tigecycline-resistant isolates may be due to the existence of *MarR*, *SoxR/SoxS*, *AcrR,* and *MexZ*. In short, the MICs of the isolates we tested for 14 antibiotics supported the presence of drug-resistance genes in these isolates as well as the existence of MICs for these antibiotics.

Among the drug resistance genes mentioned earlier in this article, *gyrA* and *ParC* are genes encoding DNA helicase and topoisomerase IV in cells, and the latter two are the target sites of quinolone drugs [31]. Mutations in *gyrA* and *ParC* can change the target sites, making the drugs unrecognizable, thus leading to the formation of drug resistance [32]. *GlpT* and *UhpT* are the transport proteins of fosfomycin, and they are also symporters of glycerol-3-phosphate and glucose-6-phosphate. When *GlpT* and *UhpT* are mutated, fosfomycin cannot be transported into the cell, resulting in a significant decrease in cell sensitivity to fosfomycin [33,34]. *FtsI* encodes penicillin-binding protein 3 (PBP3), which is the active site of beta-lactam. Mutations in FtsI make drugs unrecognizable [35]. *MarR* represses the transcription of *MarRAB* by binding to *MarO* and negatively controlling the *MarA*-dependent expression of other genes in the regulon [36]. By mutation of *MarR* or *MarO*, the repressor is rendered inactive. The resulting overexpression of *MarA* produces antibiotic resistance by increasing the expression of the major multidrug efflux pump *AcrAB-TolC* and down-regulating the outer membrane protein *OmpF* via the small RNA (sRNA) *MicF* [37]. *SoxR/S* is a chromosomally encoded genetic system that confronts low-level MDR in *E. coli* and *S. Typhimurium* [38]. The single point mutations or other unknown changes of *SoxR* lead to the high expression of *SoxS*, which can increase efflux pump activity and decrease cell permeability, creating resistance to a variety of antibiotics [39]. *AcrR* is an HTH-type transcriptional regulator, a local transcriptional inhibitor, which can inhibit the transcription of the *acrB* gene, which encodes multi-drug efflux pump *acrB*. When point mutation occurs in *acrR*, it loses its inhibitory effect on *acrB*, resulting in the high expression of *acrB* and an increase in the number of efflux pumps [40]. *MexZ* plays a negative role in the expression of the *mexXY* efflux pump in *P. aeruginosa*. *MexXY* plays an important role in the efflux of a variety of antibiotics. *MexZ* mutants’ cloud loses the inhibition on *mexXY* which increases the number of pumps [41]. To the best of our knowledge, *MexZ_K127E* is a new point mutation of the *MexZ* gene in *P. aeruginosa* found in this study.

In addition, through the MLST analysis of 11 isolates, we found that only SX-2E-03 and SX-1E-06 belonged to the same ST (ST101). It was evident from the STs of isolates from different regions, as well as from isolates within the same region, that there is a wide genetic diversity among them. It is imperative to adopt more flexible strategies for clinical treatment and prevention because there are not only many kinds of drug-resistant bacteria in northwest China, but also many STs with different drug-resistance profiles. A number of mechanisms are thought to contribute to Gram-negative bacteria’s intrinsic and acquired drug resistance. In our data, WGS accurately identifies the exact mechanism of antibiotic resistance for Gram-negative isolates.

## 5. Conclusions

According to our findings, tigecycline-resistant bacteria were found on farms in Gansu, Qinghai, Xinjiang, Sichuan, and Shaanxi in northwest China. There are many different types of multidrug-resistant STs bacteria. As a result of sequencing and analyzing the WGS of the isolates, we identified drug-resistance genes and virulence factors. A joint analysis of the drug-resistance genes and drug-resistance spectrum of the isolates also confirmed the presence of drug-resistance genes. In addition, based on epidemiological investigation and WGS analysis, despite the low resistance rate of tigecycline, we believe that the multidrug resistance of tigecycline-resistant isolates in northwest China is a serious problem; additionally, the mechanism of drug resistance is complex, which makes prevention and control more difficult. In light of this, we should carry out more research on MDR bacteria and increase surveillance of these bacteria.

## Figures and Tables

**Figure 1 microorganisms-10-02432-f001:**
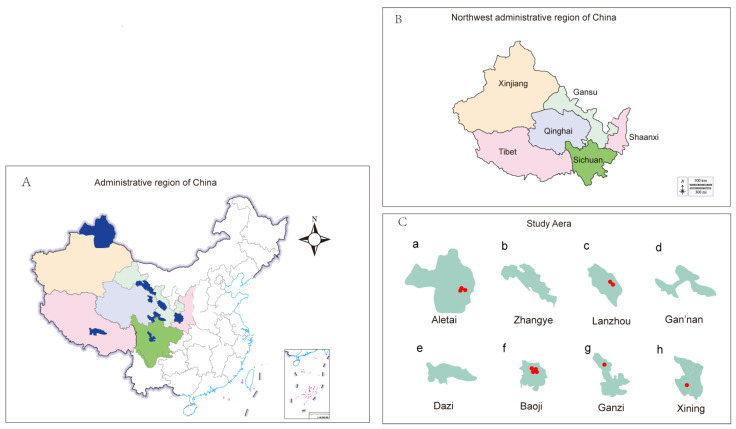
Map of the study plots showing the distribution of tigecycline-resistant isolates. (**A**) Sampling areas in the provinces in northwest China. (**B**) The provinces in northwest China are given different colors. (**C**) The study areas, where a red mark indicates a positive result.

**Figure 2 microorganisms-10-02432-f002:**
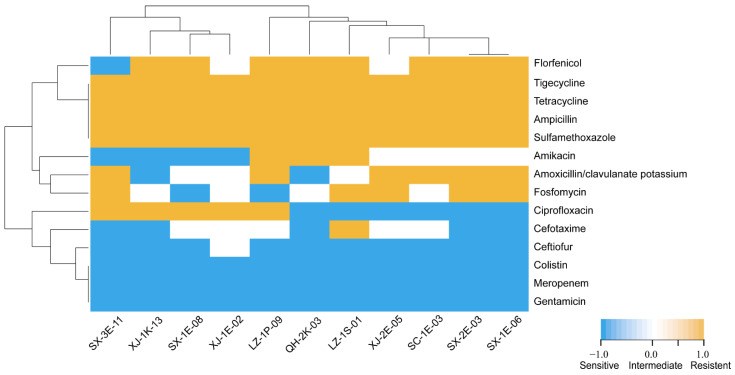
Cluster analysis of tigecycline-resistant characteristics of isolates. Vertical and horizontal trees represent clustering relationships. The ordinate is the antibiotic, and the abscissa is a tigecycline-resistant isolate. Notes (antibiotic resistance pattern): SX-1E-06: AMP-AMC-SXT-TIG-TET-FFC-FOS; SX-2E-03: AMP-AMC-SXT-TIG-TET-FFC-FOS; SC-1E-03: AMP-AMC-SXT-TIG-TET-FFC; XJ-2E-05: AMP-AMC-SXT-TIG-TET-FOS; LZ-1S-01: AMP-CTX-AMK-SXT-TIG-TET-FFC-FOS; QH-2K-03: AMP-AMK-SXT-TIG-TET-FFC; LZ-1P-09: AMP-AMC-AMK-CIP-SXT-TIG-TET-FFC; XJ-1E-02: AMP-CIP-SXT-TIG-TET; SX-1E-08: AMP-CIP-SXT-TIG-TET-FFC; XJ-1K-13: AMP-CIP-SXT-TIG-TET-FFC; and SX-3E-11: AMP-AMC-CIP-SXT-TIG-TET-FOS.

**Figure 3 microorganisms-10-02432-f003:**
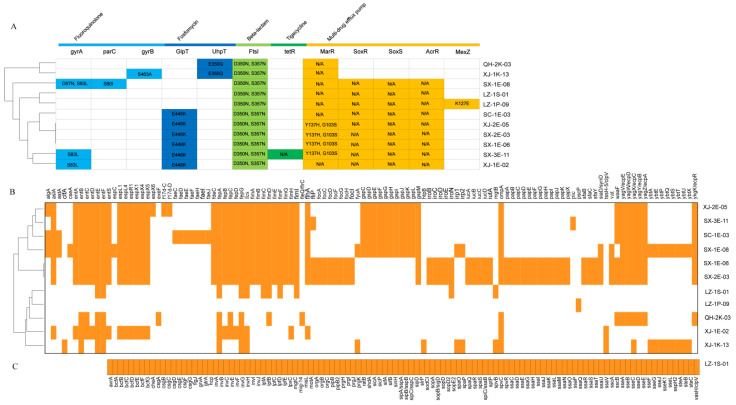
The acquired antimicrobial resistance (AMR) gene and virulence genes of tigecycline-resistant isolates with whole-genome sequences. (**A**) Various types of AMR genes are labeled with different colors. White squares indicate no AMR genes. (**B**) Cluster analysis of common virus gene; (**C**) LZ-1S-09 specific virulence gene profiles. Colored squares represent virulence genes, and white squares indicate no virulence gene.

**Table 1 microorganisms-10-02432-t001:** Geographic distribution of the samples collected in this study.

Region	Study Area	Isolates, n 2021 March–June	2021 September–December	2022 March–June	2022 Seprember–December	Total	Prevalence
Xinjiang	Aletai	60	-	-	-	60	3/60, 5.0%
Gansu	Zhangye	30	-	-	-	30	0
	Lanzhou	-	-	-	30	30	2/30, 6.67%
	Gan’nan	-	-	30	-	30	0
Shaanxi	Baoji	-	30	30	-	60	4/60, 6.67%
Sichuan	Ganzi	-	-	-	30	30	1/30, 3.33%
Tibet	Dazi	30	-	-	-	30	0
Qinghai	Xining	-	30	-	-	30	1/30, 3.33%

**Table 2 microorganisms-10-02432-t002:** Resistance pattern of tigecycline-resistant isolates against selected antimicrobial agents.

Antimicrobial Agents	Bacterial Isolates (MIC µg/mL) ^a^										
	SX-3E-11	SX-2E-03	SX-1E-06	SX-1E-08	SC-1E-03	XJ-2E-05	XJ-1E-02	LZ-1S-01	LZ-1P-09	QH-2K-03	XJ-1K-13
Ampicillin	>256	>256	>256	>256	>256	>256	>256	>256	>256	>256	>256
Amoxicillin/clavulanate potassium	32	32	32	16	32	32	16	16	16	2	4
Gentamicin	0.5	2	0.5	2	8	4	32	4	32	1	8
Cefotaxime	1	4	2	16	32	32	16	>256	32	4	2
Meropenem	0.25	0.25	0.125	0.125	0.5	1	1	1	0.5	0.25	0.125
Amikacin	16	32	32	16	32	32	16	128	32	64	8
Ceftiofur	2	0.5	1	4	0.25	0.5	0.25	0.5	0.5	0.5	0.25
Colistin	0.5	0.125	0.25	0.25	0.125	0.125	0.125	0.125	0.5	0.125	0.125
Ciprofloxacin	8	0.125	0.0625	8	0.125	0.0625	8	0.0625	8	0.125	8
Sulfamethoxazole	>256	>256	>256	>256	>256	>256	>256	>256	>256	>256	>256
Tetracycline	16	64	64	32	8	16	64	64	64	8	64
Tigecycline	8	4	8	8	8	4	8	8	8	32	32
Florfenicol	0.25	64	64	64	32	8	8	64	8	64	64
Fosfomycin	16	16	16	2	8	16	8	16	2	8	8

^a^: Minimum inhibitory concentration breakpoints.

**Table 3 microorganisms-10-02432-t003:** The summary statistics of the assembled draft genomes of tigecycline-resistant isolates.

Genomic Data	SX-3E-11	SX-2E-03	SX-1E-06	SX-1E-08	QH-2K-03	SC-1E-03	XJ-2E-05	XJ-1K-13	XJ-1E-02	LZ-1P-09	LZ-1S-01
Raw_data	3,464,897,100	3,668,641,500	2,981,649,300	3,394,535,100	3,267,123,600	3,229,710,600	2,799,241,800	3,049,228,200	3,552,870,300	1,913,231,700	3,959,209,500
Raw reeds	23,099,314	24,457,610	19,877,662	22,630,234	21,780,824	21,531,404	18,661,612	20,328,188	23,685,802	12,754,878	26,394,730
Read_length	150	150	150	150	150	150	150	150	150	150	150
Sequence length	5,154,027	5,417,711	5,417,297	5,071,184	5,574,832	5,105,116	5,066,943	6,028,924	4,819,548	31,174,726	4,754,898
Scaffolds count	127	155	152	104	81	187	129	82	188	364	31
% GC ^a^	50.98	50.63	50.38	50.54	56.98	50.22	50.53	55.05	51.45	56.71	52
Q20(%) ^b^	98.01	97.99	97.98	97.95	97.9	97.96	97.8	98.04	97.8	97.73	97.89
Largest contig	626,493	387,573	387,573	593,011	905,104	402,856	556,679	465,690	231,092	556,019	773,785
N50 ^c^	181,274	123,205	123,205	143,900	234,490	111,759	177,264	221,920	96,022	159,433	467,849
N90 ^d^	32,857	30,785	31,761	39,038	54,273	25,327	34,748	70,423	18,581	57,075	152,566
L50 ^e^	9	14	14	11	7	14	9	9	17	61	4
L75 ^f^	17	27	27	22	15	30	20	17	34	128	7

^a^: DNA (G+C) mol%; ^b^: quality score, Q = −10log10(e); ^c^: scaffold N50 length; ^d^: scaffold N90 length; ^e^: Number of scaffolds L50; ^f^: Number of scaffolds L75.

**Table 4 microorganisms-10-02432-t004:** ST and allele of tigecycline-resistant isolates.

Isolate Name	Allele adK	fumC	gyrB	icd	mdh	purA	recA	ST ^a^	CC ^b^
SX-3E-11	6	6	15	56	11	26	6	2144	-
SX-2E-03	43	41	15	18	11	7	6	101	ST101 Cplx
SX-1E-06	43	41	15	18	11	7	6	101	ST101 Cplx
SX-1E-08	10	11	4	8	179	8	2	3944	-
SC-1E-03	6	4	1323	1	9	2	7	13667	-
XJ-2E-05	6	4	4	16	24	8	14	58	ST155 Cplx
XJ-1E-02	6	107	1	95	69	8	20	536	ST399 Cplx
	gapA	infB	mdh	pgi	phoE	rpoB	tonB	ST	
QH-2K-03	2	1	2	1	10	1	12	999	-
XJ-1K-13	1	7	2	30	4	1	2	65	-
	acsA	aroE	guaA	mutL	nuoD	ppsA	trpE	ST	
LZ-1P-09	24	16	20	16	15	14	21	68	-
	aroC	dnaN	hemD	hisD	purE	sucA	thrA	ST	
LZ-1S-01	5	2	3	7	31	41	11	92	-

^a^: Sequence typing; ^b^: Clonal complex.

## Data Availability

Genomic sequences of *E. coli*, *K. pneumoniae*, *S. Typhimurium,* and *P. aeruginosa* isolates have been deposited in the BioProject database. BioSample accessions include SAMN31178477, SAMN31203371, SAMN31203373, SAMN31204682, SAMN31205078, SAMN31205552, and SAMN31205553. Genomic sequences of *K. pneumoniae* isolates have been deposited in the BioProject database. BioSample accessions include SAMN31205649 and SAMN31205656. Genomic sequences of *P. aeruginosa* isolates have been deposited in the BioProject database. BioSample accessions include SAMN31205657. Genomic sequences of *S. Typhimurium* isolates have been deposited in the BioProject database. BioSample accessions include SAMN31205675.

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
