# Peer review of "Antimicrobial Resistance Surveillance of Tigecycline-Resistant Strains Isolated from Herbivores in Northwest China"

_microorganisms, 2022, doi:10.3390/microorganisms10122432_

Round 1

Reviewer 1 Report

Comments

1.      Lin 13-15 please rephrase as “All isolates were multiple drug-resistant (MDR) shown resistant to more than three kinds of antibiotics.  

Introduction

2.      Is KPN the standard abbreviation for Klebsiella pneumoniae, please replace it by (K. pneumoniae) and follow in all text

3.      S. Tm?…please replace by full species name and follow in all manuscript

4.      P. Aeruginosa, write as P. aeruginosa

5.      Line 49-50, here you can also state that mcr type colistin-resistant is widely reported in Enterobacteriaceae so tigecycline is one of the last resort antibiotics for treating these superbugs. You can cite these articles for this statement (10.5812/jjm.96646;  10.1590/0037-8682-0237-2019; 10.1111/jam.15469)

Methods

6.      Please mention the brand name of antimicrobial agents

7.      Line # 89, please mention the CLSI reference

Results

8.      Line 138-147, you already mentioned these profiles in table, no need to mention them in the text, just refer to table here.

9.      In figure two please specify the similarity index

10.   Figure 3 is look like screen short of Excel sheet, please improve its quality

1.   You mention the ST, what is its clonal complex??

1.   No information about the plasmid types

1.   Please provide the figures of whole genome sequencing data and mention the location of tet gene, insertion sequence and background genes data.

Author Response

Dear reviewers

Re: Manuscript ID: 1997455 and Title: Antimicrobial resistance surveillance of Tigecycline resistant strains isolated from herbivores in Northwest China.

Thank you for your letter and the reviewers’ Comments and Suggestions concerning our manuscript. Those comments are valuable and very helpful. We have read through comments carefully and have made corrections. Based on the instructions provided in your letter, we uploaded the file of the revised manuscript.

Reviewer #1:

Q1: Lin 13-15 please rephrase as “All isolates were multiple drug-resistant (MDR) shown resistant to more than three kinds of antibiotics. 

Response:We are grateful for the suggestion. We have replaced the original sentence with the one you suggested

Q2: Is KPN the standard abbreviation for Klebsiella pneumoniae, please replace it by (K. pneumoniae) and follow in all text.

Response:Thank you for pointing out the non-standard abbreviation of the article. We have replaced KPN with K. pneumoniae.

Q3: S. Tm?…please replace by full species name and follow in all manuscript.

Response:Thank you for your thoughtful comments. We have mentioned its full specifications, Salmonella typhimurium, in lin 36. In addition, all S. Tm is replaced by S. typhimurium

Q4:  P. Aeruginosa, write as P. aeruginosa

Response:Thank you for pointing out the non-standard abbreviation of the article. We have replaced P. Aeruginosa with P. aeruginosa.

Q5:Line 49-50, here you can also state that mcr type colistin-resistant is widely reported in Enterobacteriaceae so tigecycline is one of the last resort antibiotics for treating these superbugs. You can cite these articles for this statement (10.5812/jjm.96646;  10.1590/0037-8682-0237-2019; 10.1111/jam.15469)

Response:We agree with the comment and re-wrote the sentence in the revised manuscript.

Q6: Please mention the brand name of antimicrobial agents.

Response:Thank you for pointing out the missing information. We have supplemented the brand name of antimicrobial agents in the manuscript.

Q7:Line # 89, please mention the CLSI reference

Response:Thank you for pointing out the missing information. We have quoted CLSI literature in the manuscript.

Q8: Line 138-147, you already mentioned these profiles in table, no need to mention them in the text, just refer to table here.

Response:We are grateful for the suggestion. We have deleted the part that is duplicate with Table 2 in 3.2 and referred to Table 2.

Q9: In figure two please specify the similarity index.

Response:Thanks for your careful suggestion, we have added the missing information.

Q10:Figure 3 is look like screen short of Excel sheet, please improve its quality

Response:We are grateful for the suggestion. We have improved the clarity of Figure 3 and carried out cluster analysis.

Q11:You mention the ST, what is its clonal complex??

Response:We are grateful for the suggestion. We added their clonal complex in Table 4.

Q12:No information about the plasmid types

Response:Thank you for your thoughtful comments. Due to the nature of second-generation whole genome sequencing, it is difficult to pinpoint whether or not it is a plasmid sequence. For this reason, second-generation sequencing will be used to screen strains and identify targets before moving on to the next step of the experiment.

Q13:Please provide the figures of whole genome sequencing data and mention the location of tet gene, insertion sequence and background genes data

Response:Thank you for your thoughtful comments. I apologize for not being able to make the changes exactly as you suggested. In this study, second-generation sequencing was used, and all sequencing results have been uploaded. Refer to the Data Availability Statement for more information.

We would love to thank you for allowing us to resubmit a revised copy of the manuscript and we highly appreciate your time and consideration.

Sincerely.

qiwei CHEN

Reviewer 2 Report

This study aimed to investigate the genetic characteristics and variations of tigecycline-resistant gram-negative isolates from herbivores in northwest China

- You need to provide the author's affiliations.

- Figure 1 is very small.

- I do not see any information in materials and methods about bacterial isolation and identification. How did you make MIC then?

- on what bases did you select the list of antibiotics you tested?

- What about the distribution of each pathogen within your samples? you will need to add this.

- Section 3.1 needs to be rewritten and provide a table to distribute the number, pathogen, prevalence, and so on.

- Provide the abbreviation's full name in the footnote of table 2.

- Names of the pathogens need to be in italic throughout the manuscript.

- please use the same font throughout the manuscript.

- Figure 3: its better to make a heat map rather than take a caption from an excel sheet.

- I do not see any stats in your results.

- you need to provide a correlation between phenotypic and genotypic analysis.

- I expected to see better analysis for the whole genome sequence data and better figures. This part needs to be rewritten.

Author Response

Dear reviewers

Re: Manuscript ID: 1997455 and Title: Antimicrobial resistance surveillance of Tigecycline resistant strains isolated from herbivores in Northwest China.

Thank you for your letter and the reviewers’ Comments and Suggestions concerning our manuscript. Those comments are valuable and very helpful. We have read through comments carefully and have made corrections. Based on the instructions provided in your letter, we uploaded the file of the revised manuscript.

Reviewer #2:

Q1:You need to provide the author's affiliations.

Response:Thank you for pointing out the missing information. We added the author's affiliation at the beginning of the manuscript.

Q2:Figure 1 is very small

Response:We are grateful for the comment. We have improved the quality of Figure 1 and supplemented the distribution of isolates

Q3:I do not see any information in materials and methods about bacterial isolation and identification. How did you make MIC then?

Response:Thank you for your thoughtful comments. We have mentioned "We screened the tigecycline persistent strains in LB plate with 2μg/mL tigecycline[20]. Species and genus of the screened strains were identified by 16S sequencing”We have supplemented the 16s sequencing results in the supporting documents

Q4:on what bases did you select the list of antibiotics you tested?

Response:I'm sorry I didn't explain clearly why we chose these antibiotics. According to literature [22], we selected the drug sensitivity group of gram-negative atmospheric bacteria, which are various antibiotics used in clinical treatment of gram-negative bacterial infections.

Q5:What about the distribution of each pathogen within your samples? you will need to add this.

Response:Thank you for pointing out the missing information. We have added this information to Figure 1C.

Q6:Section 3.1 needs to be rewritten and provide a table to distribute the number, pathogen, prevalence, and so on.

Response:Thank you for pointing out the missing information. The specific sampling time, sampling volume, prevalence and geographical location are shown in Table 1 and Figure 1.

Q7: Provide the abbreviation's full name in the footnote of table 2. 

Response:We are grateful for the suggestion. We have added the abbreviations of their full names to the footnotes of Table 2 and Table 3

Q8: Names of the pathogens need to be in italic throughout the manuscript.

Response:Thank you for pointing out the format problem in my manuscript. I have revised the pathogen in the full text to italics

Q9:please use the same font throughout the manuscript.

Response:Thank you for pointing out the format problem in my manuscript. I have unified the fonts in the manuscript as the font requirements in the journal template

Q10:Figure 3: its better to make a heat map rather than take a caption from an excel sheet.

Response:Thank you for providing better ideas for my manuscript. I have shown Figure 3 in the form of heat map.

Q11:- I do not see any stats in your results.

Response:Thank you for pointing out the deficiencies in my manuscript. I have added the statistical results in 3.1, 3.2 and 3.4 of the results.

Q12:you need to provide a correlation between phenotypic and genotypic analysis.

Response:Thank you for your suggestion. We have added the analysis between phenotype and genotype in 3.4.

Q13:I expected to see better analysis for the whole genome sequence data and better figures. This part needs to be rewritten.

Response:Thank you very much for your valuable comments which cost us time and energy. We have improved the pictures and analysis in the result.

We would love to thank you for allowing us to resubmit a revised copy of the manuscript and we highly appreciate your time and consideration.

Sincerely.

qiwei CHEN

Round 2

Reviewer 1 Report

Accept in present form

Author Response

Dear reviewers

Re: Manuscript ID: 1997455 and Title: Antimicrobial resistance surveillance of Tigecycline resistant strains isolated from herbivores in Northwest China.

Reviewer #1:

Thank you for your approval of my manuscript. According to your evaluation, we have carefully revised the English language and style in the manuscript. Once again, thank you for your time and effort on our manuscript. I wish you a happy life and smooth work.

Sincerely.

qiwei CHEN

Reviewer 2 Report

- I wonder, why there are more authors added in the revised version.

- The authors addressed all the issues in the manuscript, however, the manuscript needs to be revised for minor English errors.

Author Response

Greetings, reviewer

Re: Manuscript ID: 1997455 and Title: Antimicrobial resistance surveillance of Tigecycline resistant strains isolated from herbivores in Northwest China.

Reviewer #2:

Thank you for your time-consuming work and apologies for any inconvenience caused. Firstly, we have submitted a request to the Journal with a written explanation of the specific reasons for the addition of authors. Also, the process of adding authors was unanimously approved and signed by all co-authors. The main reason for adding new authors was to make a contribution to the article. As follows, constructive modifications were suggested and participated in the revision of the article, contributed to the framework and content of the article, and participated in the analysis of the data summary of the article.

In addition, according to your suggestion, we have corrected the English language errors in the manuscript. Thank you very much for your valuable comments on our manuscript. I wish you a happy life and smooth work.

Sincerely.

qiwei CHEN